# Dual-stimuli responsive and reversibly activatable theranostic nanoprobe for precision tumor-targeting and fluorescence-guided photothermal therapy

Xu Zhao[1], Cheng-Xiong Yang[1], Li-Gong Chen[2,3] & Xiu-Ping Yan[1,2]

The integrated functions of diagnostics and therapeutics make theranostics great potential for personalized medicine. Stimulus-responsive therapy allows spatial control of therapeutic effect only in the site of interest, and offers promising opportunities for imaging-guided precision therapy. However, the imaging strategies in previous stimulus-responsive therapies are 'always on' or irreversible 'turn on' modality, resulting in poor signal-to-noise ratios or even 'false positive' results. Here we show the design of dual-stimuli-responsive and reversibly activatable nanoprobe for precision tumour-targeting and fluorescence-guided photothermal therapy. We fabricate the nanoprobe from asymmetric cyanine and glycosyl-functionalized gold nanorods (AuNRs) with matrix metalloproteinases (MMPs)-specific peptide as a linker to achieve MMPs/pH synergistic and pH reversible activation. The unique activation and glycosyl targetibility makes the nanoprobe bright only in tumour sites with negligible background, while AuNRs and asymmetric cyanine give synergistic photothermal effect. This work paves the way to designing efficient nanoprobes for precision theranostics.

[1] College of Chemistry, Research Center for Analytical Science, State Key Laboratory of Medicinal Chemical Biology, Tianjin Key Laboratory of Molecular Recognition and Biosensing, Nankai University, Tianjin 300071, China. [2] Collaborative Innovation Center of Chemical Science and Engineering (Tianjin), Tianjin 300071, China. [3] School of Chemical Engineering and Technology, Tianjin University, Tianjin 300072, China. Correspondence and requests for materials should be addressed to X.-P.Y. (email: xpyan@nankai.edu.cn).

Theranostics, which integrates both diagnostic and therapeutic functions into a single platform, has potential to propel the biomedical field towards personalized medicine[1–4]. As conventional chemotherapy often shows poor tumour specificity and suffers serious toxic effects on cancer patients[5–7], further clinical significance of 'theranostics' via simply integrating imaging probes and chemotherapeutic drugs to a single system remains to be a question under debate[8,9]. The emergence of physical stimulus-responsive therapy brings new chances and space for theranostics. Stimulus-responsive therapies are localized and relatively safe treatments, usually non-toxic themselves and allow spatial control of the therapeutic effect only in the site of interest by external stimuli such as light[1,10], magnetic field[8,11], X-ray[8], radiofrequency[12] and ultra-sound[13]. Therefore, tumour-targeting accumulation and precision imaging are essential for oncotherapy to guide the physical stimulus in the accurate location and the best of times for the most effective and little side-effect therapy. To this goal, great efforts have been devoted to conjugate various imaging techniques with therapy agents to realize imaging-guided precision therapy[14–19]. In particular,

activatable imaging modality has received increasing attention in the fabrication of theranostic agents owing to its high specificity and sensitivity[20]. Recently, several multifunctional nanocomposites have been developed to make activatable imaging-guided therapy possible with amplified imaging signals, demonstrating the unique superiority and the great potential of activatable imaging strategy for precision imaging-guided therapy[21–25]. However, to our knowledge, the signal activation in previous theranostic studies[8,21–24] is exclusively irreversible, leading to poor signal-to-noise ratio or even 'false positive' results as the activated signals are 'always on'. Hence, a paramount challenge still remains in designing smart targeted imaging-guided theranostic platforms that are capable of intelligent recognition and specific therapy of tumour tissues.

Tumour microenvironment, especially the acidic microenvironment, is a universal feature of solid tumours, regardless of tumour types or development stages[26–28]. This characteristic enables the development of 'smart' probes, which switch on only in the tumour tissue[29–32]. However, some acidic materials or other external acid stimulation may potentially pose the issue of

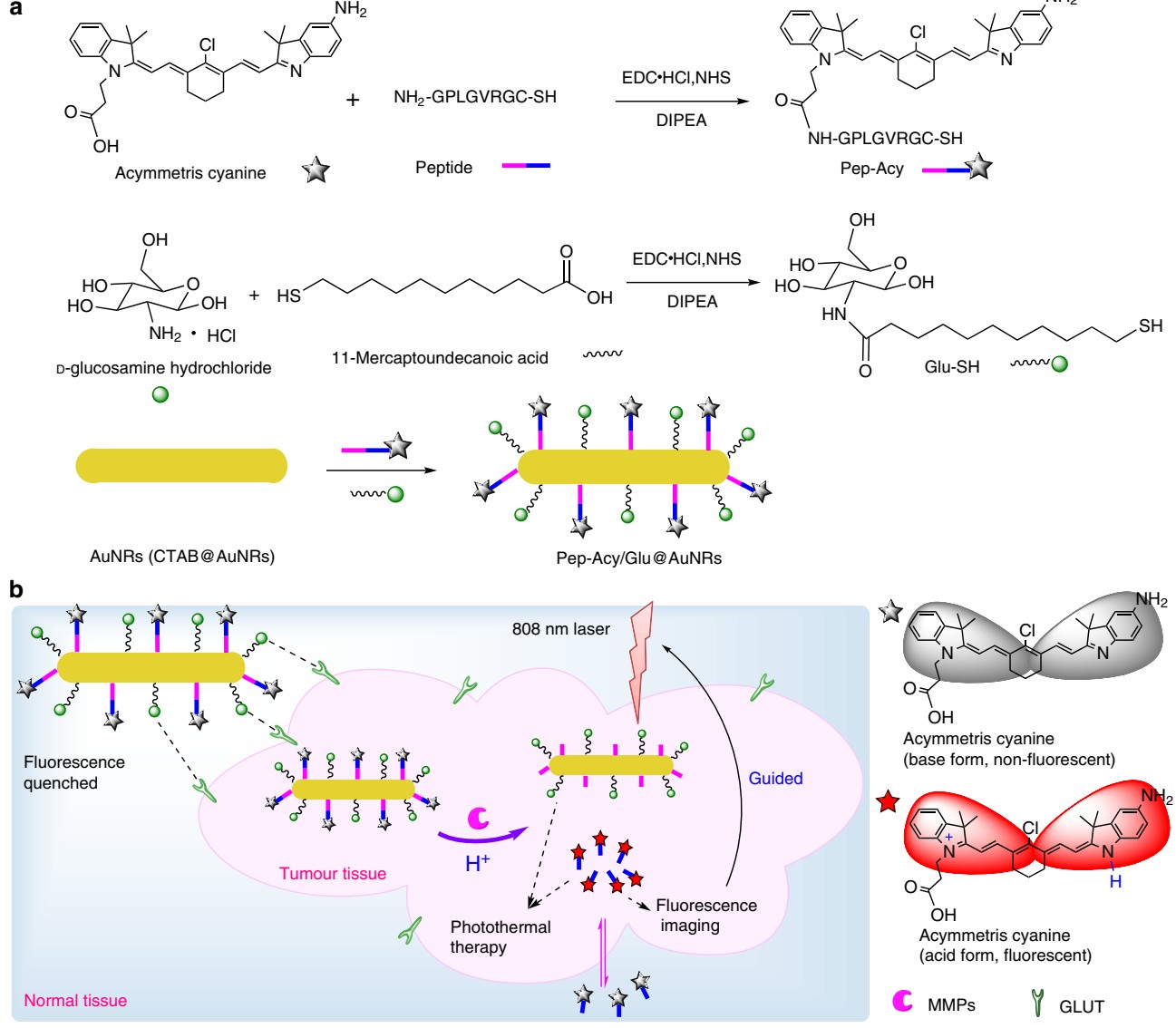

**Figure 1 | Synthesis and functionalization of the theranostic nanoprobe.** (**a**) Schematic representation of the design strategy and synthesis route. (**b**) Illustration of the nanoprobe as a MMPs/pH dual-stimuli synergistically and pH reversibly activated theranostic platform for *in vivo* tumour-targeted precision imaging-guided photothermal therapy.

nonspecific activation and cause 'false positive' result and poor signal-to-noise ratio. Therefore, more specific activated strategies, which do not merely rely on the acidic microenvironment to distinguish tumour tissue from normal tissue, are required. Matrix metalloproteinases (MMPs), overexpressed in cancer area, not only play distinct role in tumour invasiveness, metastasis and angiogenesis, but also influence multiple signalling pathways in tumour microenvironment[33–36]. Besides, D-glucose transported protein (GLUT), one of important nutrient transporters, presents particularly high-concentration in cancer cells[37–39]. Hence, MMPs and GLUT have become significant targets for *in vivo* tumour-targeted imaging and therapy[36,40,41].

Herein, we report a strategy to design and fabricate dual-stimuli synergistically and reversibly activatable multifunctional nanoprobes for *in vivo* tumour-targeting and specific imaging-guided precision photothermal therapy. We use gold nanorods (AuNRs) and asymmetric cyanine as two building units for the theranostic probe. Asymmetric cyanine serves as both the tumour-specific imaging probe and auxiliary photothermal agent due to its reversible pH-responsive near-infrared absorption and fluorescence[42]. AuNRs acts as both photothermal therapy agent and ultra-efficient quencher owing to the high-photothermal conversion efficiency and strong near-infrared absorption[22,43]. Meanwhile, a MMPs-specific peptide sequence (H₂N– GPLGVRGC–SH) serves as the linker between the AuNRs and the asymmetric cyanine near-infrared probe to build MMPs/pH synergistically and reversibly activatable theranostic nanoprobe. Further conjugation of glycosyl endows the nanoprobe active tumour-targeting ability and good biocompatibility. The as-prepared theranostic nanoprobe not only exhibits precision tumour-targeted imaging with ultra-high specificity and negligible background, but also possesses ultra-strong photothermal effect without obvious side-effect, holding great promising for theranostic application.

## Results

**Design and characterization of the theranostic probe.** The design and fabrication of our dual-stimuli synergistically and reversibly activatable multifunctional nanoprobe for tumour-targeting and specific imaging-guided precision photothermal therapy is illustrated in Fig. 1. The cetyltrimethylammonium bromide capped AuNRs (CTAB@AuNRs) were prepared according to Wang *et al.*[43] with small modification, while the asymmetric cyanine was synthesized based on our previous work[42]. To realize tumour microenvironment (for example, MMPs) activatable Förster resonance energy transfer (FRET) from the asymmetric cyanine to the AuNRs, we used a peptide sequence, H₂N–GPLGVRGC–SH (PLGVR is the cleavable site[22,40]), specifically hydrolyzable with multiple types of

MMPs, as the linker. We modified the asymmetric cyanine with the peptide H₂N–GPLGVRGC–SH via an acylation reaction between the –NH₂ group of the peptide and the –COOH group of the asymmetric cyanine with EDC·HCl and NHS as the catalyst (Supplementary Fig. 1). The prepared peptide modified asymmetric cyanine (Pep-Acy) provided a terminal-SH group to further conjugate the AuNRs via the Au–S bond to obtain a MMPs/pH dual-stimuli responsive and pH reversibly activated multifunctional nanoprobe (Pep-Acy@AuNRs). To improve the biocompatibility and to endow the active tumour-targeting ability, we further functionalized the Pep-Acy@AuNRs with thiol-terminated glucosamine (HS-Glu) via the Au–S bond to give the final multifunctional nanoprobe (Pep-Acy/Glu@AuNRs) (Fig. 1a). We also obtained HS-Glu from D-glucosamine hydrochloride and 11-mercaptoundecanoic acid via an acylation reaction (Supplementary Fig. 2). As a result, we prepared dual-stimuli synergistically and reversibly activatable multifunctional nanoprobe for *in vivo* tumour-targeting and specific imaging-guided precision photothermal therapy.

We then characterized the prepared Pep-Acy/Glu@AuNRs by transmission electron microscopy (TEM), Zeta potential analysis, dynamic light scattering, Fourier transform infrared (FT-IR), ultraviolet–vis–near-infrared absorption (UV-vis-NIR) and fluorescence spectroscopy. The as-prepared Pep-Acy/Glu@AuNRs presents well-dispersed and homogenous core–shell structure (Fig. 2a). The inside core AuNRs has a dimension of (50.1 ± 2.3) nm × (11.1 ± 0.9) nm, while the outer shell is a uniform thin grey layer with a thickness of ca. 1.6 nm due to the presence of Pep-Acy and Glu-SH (Fig. 2a cf. Supplementary Fig. 3a,b). Conjugation of Pep-Acy and Glu-SH remarkably increased the hydrodynamic size of the AuNRs from 58 to 149 nm due to the large hydrodynamic volume of glycosyl[44] (Supplementary Fig. 4a), and turned the Zeta potential from + 66.4 into − 1.1 mV (Supplementary Fig. 4b) as the initial CTAB with abundant positive charge was exchanged by Pep-Acy and Glu-SH. The disappearance of the stretching vibration band of the trimethylammonium group of CTAB (around 1,471 cm⁻¹), and the presence of new characteristic peaks of the –CONH– stretching vibration of Pep–Acy and the –CH–O– CH– vibration of Glu-SH (ca. 1,649 and 1,089 cm⁻¹) in the FT-IR spectra of Pep-Acy/Glu@AuNRs confirm the successful functionalization of Pep-Acy and Glu (Supplementary Fig. 4c).

The prepared Pep-Acy/Glu@AuNRs gave two characteristic absorption bands in UV-vis-NIR spectra at pH 7.4 (Fig. 2b). One absorption band at ca. 810 nm originated from the longitudinal absorption of AuNRs and its intensity and position showed no obvious change in comparison with AuNRs, indicating no detectable aggregation happened[45–47] (Fig. 2b cf. Supplementary Fig. 3e). The other absorption band at ca. 520 nm is stronger than that of CTAB@AuNRs due to the overlap of the transverse

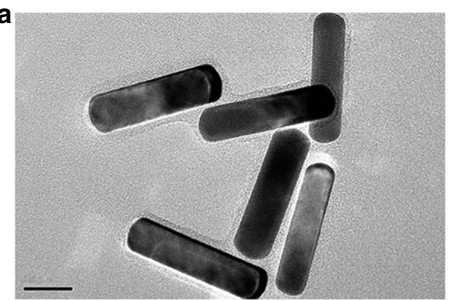

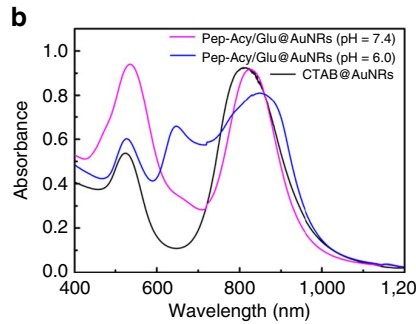

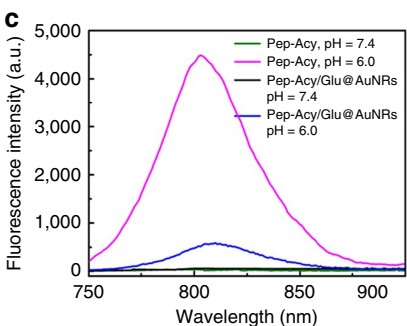

**Figure 2 | Characterization of the functionalized AuNRs. (a)** TEM images of Pep-Acy/Glu@AuNRs (Scale bar, 20 nm). **(b)** UV-vis-NIR spectra of Pep-Acy/Glu@AuNRs and CTAB@AuNRs. **(c)** Fluorescence spectra of Pep-Acy and Pep-Acy/Glu@AuNRs.

absorption of AuNRs and the absorption of Pep-Acy in its base form (Fig. 2b cf. Supplementary Fig. 5). Change of pH from 7.4 to 6.0 not only depressed the absorption band at ca. 520 nm and broadened the absorption band at 810 nm, but also produced a new absorption band at ca. 659 nm due to the transition of the Pep-Acy from its base to acid form (Fig. 2b cf. Supplementary Fig. 5). The above results also indicate that the conjugation of Pep-Acy and Glu-SH did not affect the inherent absorption properties of AuNRs and Pep-Acy. More importantly, the prepared Pep-Acy/Glu@AuNRs offered excellent colloidal stability in a complex physiological medium, Dulbecco's modified Eagle's medium (DMEM) with 10% fetal bovine serum (FBS), due to no remarkable variation in the absorption intensity and peak position in the UV-vis-NIR spectra (Supplementary Fig. 6a). Meanwhile, neither morphological changes nor aggregation was seen in the TEM images (Supplementary Fig. 6b cf. Fig. 2a). Ultraviolet–vis titration revealed about 6,234 molecules of Pep-Acy bound to each AuNR (Supplementary Methods; Supplementary Figs 7 and 8).

Conjugation of Pep-Acy to AuNRs led to prominent fluorescence quenching of Pep-Acy (Fig. 2c) due to the multi-quenching effects of Pep-Acy self-quenching and the FRET effect resulting from the overlap between the fluorescence emission of Pep-Acy (Fig. 2c) and the strong absorption of AuNRs (Fig. 2b). The fluorescence quenching efficiency was calculated to be 99.3% (Supplementary Methods), resulting in a fluorescence silent nanoprobe.

**Dual-stimuli responsive and reversible activation**. We then investigated the feasibility of multiple types of MMPs, such as MMP-2, MMP-3, MMP-7, MMP-9 and MMP-13 and/or pH stimulated fluorescence activation of Pep-Acy/Glu@AuNRs (Fig. 3a, Supplementary Fig. 9a). Incubation of Pep-Acy/Glu@AuNRs and various types of MMPs with simultaneous adjusting pH to 6.0 (simulated tumour acidic microenvironment) activated the near-infrared fluorescence. In particular, MMP-13 induced the most significant near-infrared fluorescence signal (Supplementary Fig. 9a). The fluorescence de-quenching of the nanoprobe was attributed to the detachment of Acy from AuNRs because the peptide was enzymolyzed by MMPs to inhibit the FRET process, while the acidic pH activated the detached Acy into its acidic fluorescence form. MMP-13 was therefore chosen as a typical MMP for further *in vitro* test due to its high sensitivity. The fluorescence signal recovery of Pep-Acy/Glu@AuNRs was closely correlated with the concentration of MMP-13 in an acidic microenvironment (Supplementary Fig. 9b). In contrast, no fluorescence of Pep-Acy/Glu@AuNRs was lightened up with MMP-13 in weak basic medium (pH 7.4, normal biological fluid pH) although the peptide was enzymolyzed in the presence of MMP-13 because the detached Acy was in its basic non-fluorescent form. Moreover, change of pH from weak base to weak acid (for example, pH 7.4–6.0) alone (in the absence of MMP-13) did not result in significant fluorescence signal due to the FRET process induced fluorescence quenching between Pep-Acy and AuNRs though the Acy was activated into its acidic fluorescence form. For the same reasons, the addition of MMP-13 inhibitor led to a similar result even in the presence of MMP-13 and an acidic environment (pH 6.0) (Fig. 3a). The above results indicate that the activation of Pep-Acy/Glu@AuNRs should be carried out with synergetic stimulation of acidic microenvironment and MMP-13, ensuring its ultra-high specificity. Importantly, the activation of Pep-Acy/Glu@AuNRs is reversible with pH change (Supplementary Fig. 10). The unique features of MMP/pH dual-stimuli-responsive and pH reversible activation enable

Pep-Acy/Glu@AuNRs for precision tumour-specific imaging with negligible background signals and no 'false positive' results.

**Tumour-targeting imaging and biodistribution**. The above results encouraged us to apply the Pep-Acy/Glu@AuNRs for cell imaging. Both MMPs positive (murine squamous cell carcinoma cells, SCC-7) and negative cells (human embryonic kidney transformed cells, 293T) were employed as model cells. The expression levels of MMPs including MMP-2, MMP-9 and MMP-13 in SCC-7 cells are much higher than those in 293T cells, as verified by western blotting and quantitative MMPs ELISA (Supplementary Fig. 11). We first evaluated the cytotoxicity of the Pep-Acy/Glu@AuNRs towards both SCC-7 (Fig. 3b) and 293T cells (Fig. 3c). Pep-Acy/Glu@AuNRs exhibited no significant cytotoxicity for the studied cells until the test concentration up to 60 μg ml$^{-1}$ (a high-viability of ca. 89%), slightly higher than Pep-Acy/@AuNRs (ca. 80%), whereas the CTAB@AuNRs showed remarkable cytotoxicity even at much lower concentrations. These results demonstrate that the modification strategy in our design greatly improved the biocompatibility of AuNRs.

We then studied the cell internalization and the imaging performance of Pep-Acy/Glu@AuNRs in living cells. SCC-7 cells incubated with Pep-Acy/Glu@AuNRs at pH 6.0 displayed remarkable fluorescence, about 2.7-fold increase of mean fluorescence intensity (MFI) compared to that in glucosamine pre-treated SCC-7 cells (GLUT receptor was blocked) or the SCC-7 cells incubated with Pep-Acy@AuNRs although the same concentration of nanoprobe (as Au) was applied (Fig. 3d,e). Besides, the intracellular Au content in the cells treated with Pep-Acy/Glu@AuNRs also showed about 2.5-fold increase compared with the above controls (Supplementary Fig. 12). These results indicate that Pep-Acy/Glu@AuNRs were successfully internalized into the SCC-7 cells and the cell internalization was obviously enhanced by glycosyl due to the dual-targeting effects of both enhanced permeation and retention (EPR) effect and GLUT receptor mediated active tumour-targeting effect. In comparison, the low MMPs expressed SCC-7 cells pretreated with MMPs inhibitor (Supplementary Fig. 11) showed little fluorescence, while MMPs negative 293T cells exhibited almost no fluorescence at pH 6.0 although the same amount of Pep-Acy/Glu@AuNRs was applied, showing a strong MMPs dependent fluorescence activation (Fig. 3f). On the other hand, neither SCC-7 nor 293T cells lighted up the fluorescence of Pep-Acy/Glu@AuNRs at pH 7.4, indicating the fluorescence activation was also pH dependent. Furthermore, the activated fluorescence signal in SCC-7 cells at pH 6.0 disappeared as pH turned to 7.4. These results clearly indicate that the fluorescence of Pep-Acy/Glu@AuNRs could be switched on if and only if acidic microenvironment and overexpressed MMPs co-exist, making it promising for intelligent tumour-specific imaging. Besides, the imaging specificity of Pep-Acy/Glu@AuNRs towards pH and MMPs was confirmed and quantified by flow cytometry analysis as well (Supplementary Figs 13 and 14).

The inspiring results of the *in vitro* MMP/pH synergistic fluorescence activation and cell imaging encouraged us to investigate whether Pep-Acy/Glu@AuNRs is good for *in vivo* precision tumour-targeting imaging (Fig. 4; Supplementary Fig. 15). For this purpose, we randomly divided nude mice with two subcutaneous SCC-7 tumours in two sides of the groin (that is, L-tumour and R-tumour) into two groups (Group A and B). For both Group A and B, a dramatic fluorescence signal appeared in the untreated R-tumour from 2 h after the intravenous injection of Pep-Acy/Glu@AuNRs. The fluorescence signal gradually increased with the maximum at ca. 4 h, indicating 4 h is the most appropriate time point for the follow-up *in vivo*

**Figure 3 | *In vitro* evaluation of Pep-Acy/Glu@AuNRs.** (**a**) Responses of the fluorescence spectra of Pep-Acy/Glu@AuNRs to MMP-13 and pH with or without inhibitor. (**b,c**) Cells cytotoxicity of CTAB@AuNRs, Pep-Acy@AuNRs and Pep-Acy/Glu@AuNRs against SCC-7 and 293T cells, respectively ($n = 5$, *$P < 0.05$). (**d**) Flow cytometry analysis. Inset: Corresponding mean fluorescent intensity (MFI). Cyan and magenta: SCC-7 cells treated with Pep-Acy/Glu@AuNRs (cyan) and Pep-Acy@AuNRs (magenta), respectively. Blue: Glucosamine per-blocked SCC-7 cells treated with Pep-Acy/Glu@AuNRs. Black: SCC-7 cells without treatment. (**e**) Cell internalization of Pep-Acy/Glu@AuNRs towards SCC-7 (Scale bar, 10 μm). (**f**) Cell imaging of Pep-Acy/Glu@AuNRs towards SCC-7, 293T and inhibitor pretreated SCC-7 cells (Scale bar, 10 μm). Center values and error bars are defined as mean and s.d., respectively. Statistical significance is assessed by a two-factor analysis of variance (two-way ANOVA).

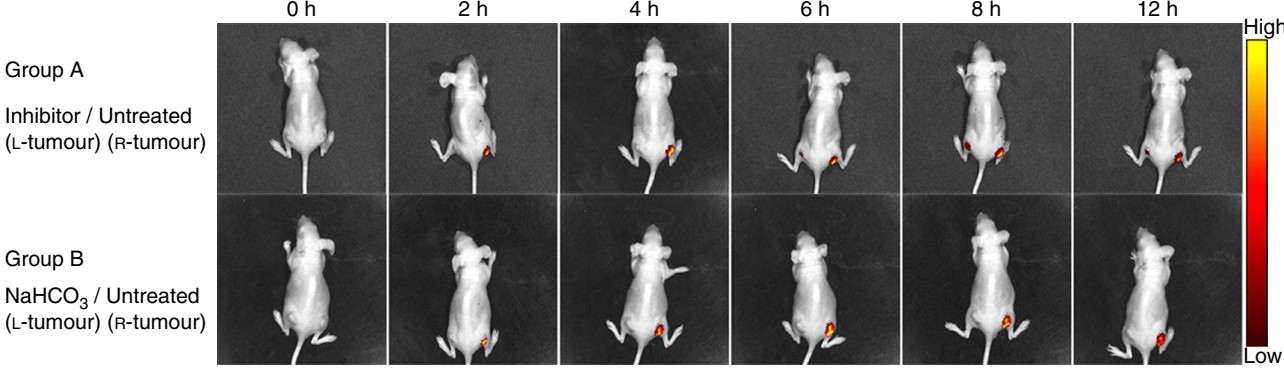

**Figure 4 | Pep-Acy/Glu@AuNRs-mediated *in vivo* fluorescence images in SCC-7 tumour-bearing mice.** L-tumour was pre-treated with MMPs inhibitor (Group A) or NaHCO₃ (Group B). Images were acquired using an IVIS Lumina II *in vivo* imaging system.

photothermal therapy. Besides, the fluorescence signal was clearly identified up to at least 12 h. For comparison, MMPs inhibitor (for Group A) or NaHCO₃ (for Group B) was intratumorally administered to the L-tumour 30 min before the intravenous injection of Pep-Acy/Glu@AuNRs to change the tumour microenvironment. For Group A, only a slight fluorescence signal appeared in the MMPs inhibitor-treated L-tumour after 6 h, and the fluorescence signal in L-tumour was always significantly

weaker than that of in L-tumour during the whole process, while for Group B almost no fluorescence signal appeared in the NaHCO$_3$-treated L-tumour and the normal tissues during the whole process. The above results confirm that the fluorescence activation of Pep-Acy/Glu@AuNRs was specifically governed by MMPs enzyme and acidic microenvironment simultaneously. The results also show that Pep-Acy/Glu@AuNRs has great potential to guide the stimulus to occur in the accurate location and the best time.

We further investigated the *in vivo* biodistribution of the nanoprobe in the mice (Supplementary Fig. 16). The accumulation of the nanoprobe in the spleen and liver was inevitable due to the strong phagocytosis in reticuloendothelial system[46]. However, no fluorescence signals were observed in these organs and other normal organs during the whole process, further showing that the fluorescence activation was specifically controlled by tumour microenvironment.

**Photothermal efficiency and targeted photothermal therapy.** We then evaluated the synergistically enhanced photothermal effect in tumour acidic microenvironment from the Acy and AuNRs in Pep-Acy/Glu@AuNRs. To this point, we first investigated the photothermal effect of Pep-Acy under 808 nm laser irradiation with various power densities (Fig. 5a; Supplementary Fig. 17a). The temperature of the Pep-Acy solution at pH 6.0 gradually rose on 808 nm laser irradiation, and obviously increased with laser power density. In contrast, phosphate buffered saline (PBS) and Pep-Acy at pH 7.4 did not trigger any increase of temperature even though the 808 nm laser irradiation was conducted at 0.6 w cm$^{-2}$ for 10 min. These results clearly show that the Pep-Acy activated at pH 6.0 also offered photothermal effect due to the strong near-infrared absorption of its acid form (Supplementary Fig. 5). Owing to the obvious photobleaching of Pep-Acy at 0.8 w cm$^{-2}$, 0.6 w cm$^{-2}$ or lower powers were chosen as the irradiating powers in the following experiments.

We next compared the photothermal efficiency of Pep-Acy/Glu@AuNRs and CTAB@AuNRs with the same amount of Au at pH 6.0 to reveal the synergistic photothermal effect from the Acy and AuNRs (Fig. 5b; Supplementary Fig. 17b,c). Both Pep-Acy/Glu@AuNRs and CTAB@AuNRs exhibited distinct increase of temperature on 808 nm laser irradiation and displayed a power density-dependent hyperthermia. Moreover, Pep-Acy/Glu@AuNRs gave faster heating rate than CTAB@AuNRs due to the acid-activated auxiliary photothermal effect of Pep-Acy. Besides, we further verified the photothermal effect of Pep-Acy/Glu@AuNRs in SCC-7 cells (Fig. 5c). Irradiation of Pep-Acy/Glu@AuNRs with 808 nm laser resulted in obvious inhibitory effect on the cells in a

time- and laser power density-dependent manner. These results show that Pep-Acy/Glu@AuNRs has great potential for photothermal effect.

To evaluate the photothermal ablation of Pep-Acy/Glu@AuNRs *in vivo*, we randomly divided SCC-7 tumour-bearing mice with approximately the same tumour volume into four groups (three mice each group). The treatment group refers to the mice treated with Pep-Acy/Glu@AuNRs under 808 nm laser irradiation (0.6 w cm$^{-2}$ for 10 min) at 4 h post-injection, guided by the *in vivo* imaging results, while three control groups include untreated mice, the mice injected with PBS subjected to laser exposure, and the mice injected with Pep-Acy/Glu@AuNRs without laser irradiation. On 808 nm laser irradiation, the temperature of tumours rose rapidly and reached about 50 °C within 6 min in the treatment group (Fig. 6a; Supplementary Fig. 18), which was sufficient to kill the cancer cells effectively[46]. In contrast, the temperature only slightly increased in the PBS group even under laser irradiation for 10 min (ca. 37 °C). The above results indicate that the Pep-Acy/Glu@AuNRs has effective photothermal therapeutic effect and the laser at this intensity itself has non-photothermal side-effect. To further investigate the photothermal ablation effect of Pep-Acy/Glu@AuNRs, we monitored the tumour growth rate up to 14 days (Fig. 6b,d). The tumour in the treatment group was almost suppressed without obvious reoccurrence during the period of our observation, whereas the tumour in the control groups exhibited significant increase. These results show that Pep-Acy/Glu@AuNRs offers excellent photothermal therapeutic effect. Moreover, neither death nor abnormal behaviour was observed in the treatment group during the whole therapeutic process. Besides, neither distinct body weight loss nor noticeable organ damage or inflammation was observed in the treatment group compared to the controls (Fig. 6c,e). All the above results show that the Pep-Acy/Glu@AuNRs can act as an effective photothermal agent for tumour ablation by a single-intravenous injection and one-time laser irradiation with no obvious side-effect.

**Discussion**

In this study, we have taken full advantages of intrinsic weak acid[26,29,30] and high-enzyme level[25,33,35] of tumour microenvironment, and have developed a dual-stimuli-responsive and reversibly activatable theranostic nanoprobe based on asymmertric cyanine and glycosyl functionalized AuNRs for precision tumour-targeting fluorescence imaging-guided photothermal therapy. As demonstrated in Fig. 1b, once the nanoprobe is internalized into the tumour tissue due to the active tumour-targeting ability of the glycosyl, it reacts with the

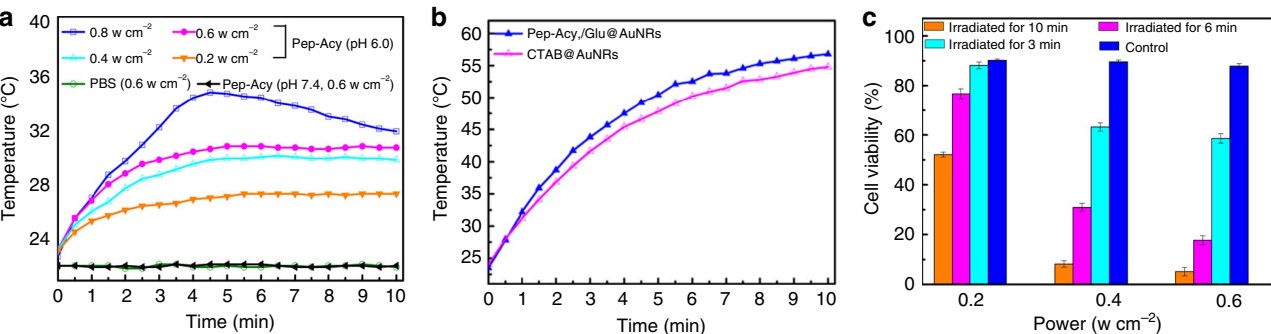

**Figure 5 | *In vitro* photothermal effects.** (**a**) Temperature change curves of Pep-Acy and PBS upon 808 nm laser irradiation. (**b**) Temperature change curves of Pep-Acy/Glu@AuNRs and CTAB@AuNRs upon 808 nm laser irradiation at 0.6 w cm$^{-2}$. (**c**) Cell viability of Pep-Acy/Glu@AuNRs with 808 nm laser irradiation against SCC-7 cells (Control: 808 nm laser irradiation (0.6 w cm$^{-2}$, 10 min) without Pep-Acy/Glu@AuNRs). Error bars are defined as s.d.

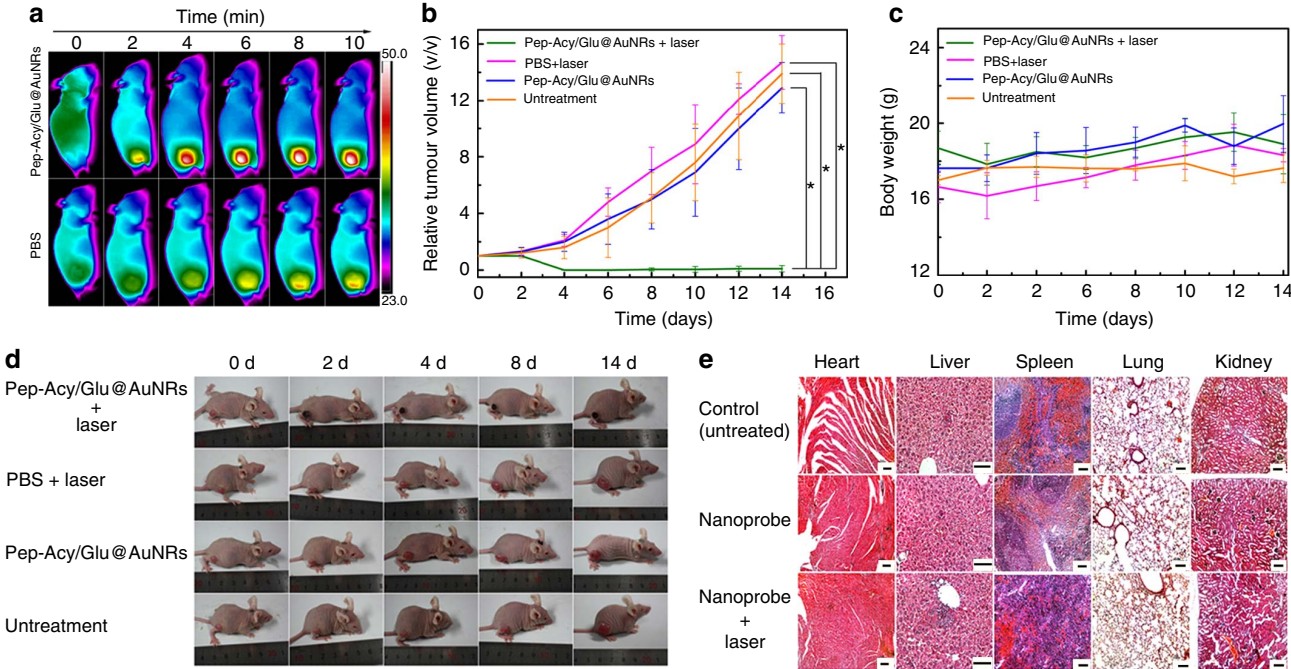

**Figure 6 | Pep-Acy/Glu@AuNRs-mediated *in vivo* photothermal therapy in SCC-7 tumour-bearing mice models.** (**a**) Representative thermal images of mice (tumour sites) subjected to 808 nm laser irradiation for 4 h after intravenous injection of Pep-Acy/Glu@AuNRs and PBS. (**b**) Relative tumour volume change of each groups of mice (*$P < 0.05$). (**c**) Body weight change curves of each groups of mice. (**d**) Representative photos of each group of mice during the whole therapy process. (**e**) H&E staining of main organs of each groups of mice after treatment (Scale bar, 100 μm). Center values and error bars are defined as mean and s.d., respectively. Statistical significance is assessed by two-way ANOVA.

tumour overexpressed MMPs in the tumour acidic microenvironment so that the peptide H₂N–GPLGVRGC–SH between Acy and AuNRs is enzymolyzed to release free Acy. The tumour acidic microenvironment makes the released free Acy present in its fluorescent acid form to produce significant near-infrared fluorescence signal. When the free Acy is transferred into normal tissue, the fluorescence of Acy is quenched as it turns into its basic non-fluorescent form, and vice versa. So, the fluorescence of the nanoprobe is lightened up only in the tumour microenviroment. Meanwhile, the released Glu@AuNRs and Acy retained in the tumour tissue serve as targeted photothermal therapy with 808 nm laser irradiation. The synergetic effect of the MMPs/pH dual-stimuli activatable nature and the reversible pH-responsive NIR fluorescence makes the Pep-Acy/Glu@AuNRs effectively overcome the shortcomings of poor signal-to-noise ratio and 'false positive' result in the previous 'always on' or irreversible 'turn-on' imaging strategies. Furthermore, such unique activation along with the double tumour-targeting ability and the enhanced photothermal effect renders the nanoprobe great potential for precision tumour-targeting and fluorescence imaging-guided high-efficiency photothermal therapy without obvious side-effect.

In conclusion, our strategy for the design of dual-stimuli-responsive and reversibly activatable theranostic nanoprobe is easily to extend to other critical players in tumour microenvironment by simply replacing the linker. For example, the use of another peptide sequence[48,49], NH₂–GGRRGGC–SH instead of H₂N–GPLGVRGC–SH, as the linker between Acy and AuNRs enables the fabrication of a nanoprobe (Pep₂-Acy@AuNRs) for specific response to cathepsin B (CtsB), a tumour-specific enzyme overexpressed in various malignant tumours[25,48,49], and the specific fluorescence activation of Pep₂-Acy@AuNRs synergistically governed by CtsB and pH (Supplementary Figs 19–23). Our strategy for the design of theranostic probes with dual-stimuli synergistically and reversibly activatable

fluorescence imaging-guided precision photothermal therapy provides a valuable approach to construct smart theranostic platform for clinical applications.

## Methods

***In vitro* enzyme and acidic microenvironment activation.** Multiple types of recombinant human MMPs (MMP-2, MMP-3, MMP-7, MMP-9 and MMP-13) (100 μg ml⁻¹) was incubated with *p*-aminophenol mercuric acid (1 mM) in TCNB buffer (50 mM Tris, 10 mM CaCl₂, 150 mM NaCl, 0.05% Brij-35 m/v, pH 7.5) at 37 °C for 2 h to obtain the activated MMPs for further use. Subsequently, various types of activated MMPs or different concentration of activated MMP-13 was added to the solution of Pep-Acy/Glu@AuNRs, respectively. After incubation at 37 °C for 2 h, the solution was adjusted to pH 6.0 to measure the fluorescence spectra. For comparison, the same amount of the activated MMP-13 solution in the presence or absence of WAY 170523 (MMP-13 inhibitor) was added to the solution of Pep-Acy/Glu@AuNRs, and incubated at 37 °C for another 2 h. Two aliquots of the resulting mixture were adjusted to pH 6.0 and 7.4 respectively to monitor the fluorescence spectra change.

**Evaluation of *in vitro* photothermal conversion efficiency.** The 0.5 ml solutions of CTAB@AuNRs, Pep-Acy/Glu@AuNRs with the same amount of Au (60 μg ml⁻¹) and Pep-Acy (1 × 10⁻⁵ M) at pH 6.0 were separately collected in 1.5 ml Eppendorf tubes, then irradiated with a 808 nm laser at various power densities (0.2, 0.4, 0.6 and 0.8 w cm⁻²) for 10 min. PBS (0.01 M) and Pep-Acy solution (1 × 10⁻⁵ M) at pH 7.4 irradiated with a 808 nm laser at 0.6 w cm⁻² for 10 min were chosen as control. The thermal images and the temperature changes of each solution were recorded real-time on a FLIR E50 thermal camera.

**Cellular experiments.** SCC-7 and 293T cells were purchased from Soochow University and Shanghai cell bank, respectively. These cells are not listed by international cell line authentication committee as cross-contaminated or mis-identified cell lines (v8.0, 2016), and not tested for mycoplasma. SCC-7 and 293T cells were cultured respectively in DMEM/high-glucose medium and Roswell park memorial institute (RPMI)-1640 medium with 10% FBS and 1% antibiotics (penicillin-streptomycin) at 37 °C under a 5% humidified atmosphere. Standard MTT assay was carried out to evaluate the cytotoxicity and *in vitro* photothermal therapy (Supplementary Methods).

To quantify the cell internalization of Pep-Acy/Glu@AuNRs, SCC-7 cells were cultured in 90 mm petri-dished until the cells reached at the end the logarithmic phase. The original medium was then replaced by fresh culture medium containing

Pep-Acy/Glu@AuNRs ($60\,\mu g\,ml^{-1}$) and incubated for an additional 24 h. After that, the cells were collected, washed twice with cold PBS and counted. Two aliquots of the cells were separately re-suspended into $500\,\mu l$ PBS and digested with $500\,\mu l$ aqua regia for flow cytometry analysis and inductively coupled plasma mass spectrometry (ICP-MS) measurement. Each experiment group was done in triplicate.

For MMPs and pH dependent cell imaging, both SCC-7 and 293T cells were seeded separately in a 12-well plate at a density of $5\times10^4$ cells per well. After incubation for 24 h, the original medium was replaced by fresh medium containing Pep-Acy/Glu@AuNRs ($60\,\mu g\,ml^{-1}$) and further incubated for 12 h. As a control to verify the specificity of MMPs enzyme, SCC-7 cells were seeded into another 12-well plate and pre-treated with MMPs inhibitor 1 h before incubating with Pep-Acy/Glu@AuNRs. The cells were then rinsed three times with PBS, fixed with 1 ml of 4% formaldehyde for 15 min, washed twice with PBS and stained with 4′,6-diamidino-2-phenylindole (DAPI) solution for 5 min. After thorough rinsing with PBS, the cells were finally soaked in PBS with pH 7.4 and 6.0 to simulate the normal biological fluid pH and tumour acidic microenvironment, respectively. The confocal fluorescence images were carried out on Leica TCS sp8 confocal laser scanning microscope. To evaluate the MMPs and pH dependent cell imaging quantitatively, flow cytometry analysis was performed in the same way.

**Animal experiments.** All animals operations were performed in compliance with the guidelines of Tianjin Committee of Use and Care of Laboratory Animals, and all experimental protocols were approved by the Animal Ethics Committee of Nankai University. Female BALB/C nude mice with 5–6 weeks old were purchased form Vital River Laboratories (Beijing, China). SCC-7 tumour-bearing mice models were established by subcutaneously injecting a suspension of SCC-7 cells ($1\times10^6$ cells in $50\,\mu l$ PBS) into the selected positions of the nude mice. In vivo imaging and photothermal therapy were carried out about 8 days later when the tumour reached about 6 mm in diameter.

For in vivo imaging, nude mice with two subcutaneous SCC-7 tumours in two sides of groin (denoted as L-tumour and R-tumour, respectively) were randomly divided into two groups (two mice each group). MMPs inhibitor or NaHCO$_3$ was intratumorally administered to the L-tumour of each experiment mouse 30 min before the intravenous injection of Pep-Acy/Glu@AuNRs ($120\,\mu g\,ml^{-1}$, $180\,\mu l$). NIR fluorescence imaging was then conducted on an IVIS Lumina II in vivo imaging system at different points in time. During the whole process, mice were anesthetized with isoflurance gas to minimize suffering. After imaging, all the mice were killed, and the tumour tissue and main organs containing lung, spleen, kidney, heart and liver were excised and digested in aqua regia for one week. The resulting solution was filtered and diluted. The content of Au in the above solution was determined by ICP-MS for monitoring the biodistribution of the nanoprobe.

For in vivo photothermal therapy, the SCC-7 tumour-bearing mice were randomly divided into four groups (three mice each group). Group A: Pep-Acy/Glu@AuNRs with 808 nm laser irradiation; Group B: PBS with 808 nm laser irradiation; Group C: Pep-Acy/Glu@AuNRs without laser irradiation; Group D: untreated. $180\,\mu l$ of $120\,\mu g\,ml^{-1}$ Pep-Acy/Glu@AuNRs in PBS was intravenously injected into the mice of Group A and C, respectively, and the same amount of PBS was injected into the mice of Group B via a tail vein. After 4 h, the mice of Group A and B were irradiated with 808 nm laser at $0.6\,w\,cm^{-2}$ for 10 min, and the temperature increase of tumour tissues was monitored by infrared thermal camera. Furthermore, the tumour volume and body weight of each mouse were recorded every two days for 14 days. The tumour volume was calculated as length $\times$ width$^2$ $\times$ 0.5, length and width are the greatest longitudinal diameter and the greatest transverse diameter, respectively, which were measured by a vernier caliper. To reflect the tumour growth intuitively, the relative tumour volume for each mouse was calculated as the tumour volume at different time/the original tumour volume before treatment. After the whole therapeutic process, all the mice were killed, the main organs including heart, liver, spleen, lung and kidney were collected and fixed with 4% formaldehyde for hematoxylin and eosin (H&E) staining in order to analyse the potential side effect of Pep-Acy/Glu@AuNRs. The H&E staining and histological sections were performed in a single-blind fashion at the Institute of Hematology and Blood Diseases Hospital, Chinese Academy of Medical Sciences, Tianjin.

**Data availability.** Data supporting the findings of this study are available within the article and its Supplementary Information files and from the corresponding author on reasonable request.

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

## Acknowledgements

X.-P.Y. thanks the support from the National Natural Science Foundation of China (Grant 21435001) and the National Basic Research Program of China (Grant No. 2015CB932001), L.-G.C. thanks the support from the National Natural Science Foundation of China (Grant 21576194) and X.Z. thanks the support from the China Postdoctoral Science Foundation (Grant 2015M581288).

## Author contributions

X.Z. developed the study concept, acquired, analysed and interpreted data, and drafted the manuscript. C.-X.Y. analysed data. L.-G.C. assisted in supervising the synthesis of Acy. X.-P.Y. wrote the manuscript, conceived the study concept, guided the project and interpreted the data.

## Additional information

**Competing interests:** The authors declare no competing interests.

**Publisher's note**: 

