## [Peer Review File · Nature Communications]

Reviewers' Comments:

Reviewer #1 (Remarks to the Author)

In this manuscript, authors described a strategy to design and fabricate dual-stimuli synergistically and reversibly activatable theranostic nanoprobes based on asymmetric cyanine and glycosyl functionalized gold nanorod (AuNRs) for in vivo tumor-targeting and specific imaging-guided precision photothermal therapy. This manuscript shows the characterization and application of a MMPs/pH dual-stimuli activatable nanoprobe (Pep-Acy/Glu@AuNRs) well. The study was done carefully and the claims support the data. Analysis appears sound and detailed. However, there are a few points that need to be addressed:

1. Fluorescence-guided photothermal therapy.

The authors claim that they demonstrated proof-of-concept for dual-stimuli synergistically and reversibly activatable fluorescence imaging-guided precision photothermal therapy.

However, in this manuscript, authors showed only photothermal therapeutic efficiency and did not demonstrate the potential of fluorescence imaging guided photothermal therapy.

In this light, the merit of reversible activation is not clear.

Authors need to discuss the merits of using this reversibly activatable nanoprobes and show the potential of fluorescence-guided photothermal therapy using these probes.

This is very important because the novelty of utilizing gold nanoparticles in both activatable probes and in PTT is low, because they have been reported a few times.

2. Discussion

Authors need to discuss more about their results in Discussion Section

3. Overall, provide statistical comparison among the group, and describe if there is a significant difference. Also, provide quantitative data of in vitro/in vivo studies.

4. Authors need to provide the MMP-13 specificity of this probe.

5. MMPs are known as secreted enzymes. Should be correlated with MMP protein expressions by western blotting in both cell lines (SCC7 and 293T).

6. In Figure 3, there are no scale bar in cellular images. And activated fluorescence signals need to be correlated with not only fluorescence intensity but also upregulated MMP-13 at protein level, e.g. by western blotting.

Reviewer #2 (Remarks to the Author)

Authors in this manuscript describe the synthesis, characterisation and evaluation at in vitro and in vivo levels of a novel dual-stimuli nanoprobe. The idea for tumor-targeting and fluorescence-guided photothermal therapy is a question that authors addresses using the new nanoprobes. The data provided could be useful for further investigation in the field. There are several points in the text referring on the importance of MMPs and how they could contribute in such proposed targeting. Apart from MMPs there are other critical players in tumour microenvironment to be addressed in such a study (for instance uPA, tPA etc). In any case a link between the matrix metalloproteinases and particularly of the types present in the cell lines used and the tumour microenvironment should be considered. For instance, types of MMPs may be activated and other not. Screening of the MMPs will be also useful.

Reviewer #3 (Remarks to the Author)

The authors reported synthesis and surface modification of a dual stimuli-responsive nanoprobe based on asymmetric cyanine and glycosyl functionalized gold nanorods (AuNRs) for both photo-thermal therapy and tumor targeting. The conjugated glucosamine was aimed to enhance both biocompatibility and tumor targeting capability of the nanoprobe while asymmetric cyanine was utilized as the tumor specific imaging probe and photo-thermal agent. The efficacy of the nanoprobe was examined using in vitro cellular experiments and in vivo animal models. I would publish this interesting paper after major revision noted below:

1. The advantages and disadvantages of the previously developed nanoprobe in the field should be mentioned in the introduction part. In the current form, the introduction is too general and does not contain key references and the last developments in the field.
2. Many of the unnecessary information (e.g., Figures 2b-d, 3b, 5, and 6b) can be transferred to the SI.
3. Figure 1 is not easy to understand and needs to be restructured.
4. It remains unknown in the text why the authors chose the peptide sequence as the linker.
5. The authors claimed that the conjugation of strong hydrophilic moieties increased the hydrodynamic size of NRs from 58 nm to 156 nm. However, the substantial reduction of the zeta potential from +66.4 into -1.1 (i.e. close to isoelectric point) suggests the formation of NRs' aggregates is responsible for the substantial increase in the size of NPs. The broad size distribution for Prep-Acy/Glu@AuNRs, in Fig. 2b, could also confirm the formation of the aggregates. This statement should be revised. Other approaches (e.g., differential centrifugal sedimentation) should be used to define the main reason of the size variations.
6. The absorption spectrum of CTAB@AuNRs should be inserted in Fig.2e for a better comparison.
7. With regards to the cytotoxicity experiments: the formation of aggregates can significantly influence the mechanism of cell internalization and the number of NRs inside the cells. This can also affect their biological identity and change the biological efficacy of these NPs. The uptake mechanism (after considering comment 5), should be defined in the revised version. Besides, I strongly recommend that the authors quantitatively compare the uptake of CTAB@AuNRs and Acy/Glu@AuNRs, to provide more tangible evidence on the substantial enhancement of NRs' biocompatibility.

POINT-BY-POINT RESPONSE TO REVIEWERS

The comments and suggestions made by the reviewers are very helpful for us to revise our manuscript. We highly appreciate the reviewers for such constructive comments. Detail reply to the comments and suggestions is made below.

Response to Reviewer 1

In this manuscript, authors described a strategy to design and fabricate dual-stimuli synergistically and reversibly activatable theranostic nanoprobe based on asymmetric cyanine and glycosyl functionalized gold nanorod (AuNRs) for in vivo tumor-targeting and specific imaging-guided precision photothermal therapy. This manuscript shows the characterization and application of a MMPs/pH dual-stimuli activatable nanoprobe (Pep-Acy/Glu@AuNRs) well. The study was done carefully and the claims support the data. Analysis appears sound and detailed. However, there are a few points that need to be addressed:

Comment 1:

Fluorescence-guided photothermal therapy.

The authors claim that they demonstrated proof-of-concept for dual-stimuli synergistically and reversibly activatable fluorescence imaging-guided precision photothermal therapy. However, in this manuscript, authors showed only photothermal therapeutic efficiency and did not demonstrated the potential of fluorescence imaging guided photothermal therapy. In this light, the merit of reversible activation is not clear.

Authors need to discuss the merits of using this reversibly activatable nanoprobe and show the potential of fluorescence-guided photothermal therapy using these probes. This is very important because the novelty of utilizing gold nanoparticles in both activatable probes and in PTT is low, because they have been reported a few times.

Reply and Corresponding Changes:

Valuable suggestion! To show the potential of the developed probe for fluorescence-guided photothermal therapy, we have fully demonstrated the dual-stimuli synergistic and reversible activation of the developed nanoprobe and its feasibility for precision tumor-targeting

fluorescence imaging both *in vitro* and *in vivo* (Please see revised manuscript, page 8, first paragraph - page 11, paragraph 2; Fig. 3a,3d-3f, Fig. 4; Supplementary Fig. 12-14,17-20 and the corresponding statements). Then, we have also revealed the photothermal therapeutic efficiency of the probe both *in vitro* and *in vivo* (Please see revised manuscript, page 11, paragraph 3 - page 13, paragraph 1; Fig. 5,6; Supplementary Fig. 21,22 and the corresponding statements). Finally, we have added one paragraph to discuss the potential of the nanoprobe for fluorescence imaging guided photothermal therapy and the merits of the probe over previous probes (Please see revised manuscript, page 13, last paragraph - page 14, first paragraph).

Comment 2:

Discussion

Authors need to discuss more about their results in Discussion Section

Reply and Corresponding Changes:

Good comments! We have extended the discussion to address the potential of the nanoprobe for fluorescence imaging guided photothermal therapy, the merits of the probe over previous probes, and the general extension of our strategy for the design of dual-stimuli-responsive and reversibly activatable theranostic nanoprobe to other critical players other than MMPs in tumor microenvironment by simply replacing the linker (Please see revised manuscript, page 13, last paragraph - page 15, first paragraph).

Comment 3:

Overall, provide statistical comparison among the group, and describe if there is a significant difference. Also, provide quantitative data of *in vitro*/*in vivo* studies.

Reply and Corresponding Changes:

Good suggestions! We have done the statistic analysis among the group and added the results in the reversed version (Please see revised manuscript, the captions for Fig. 3b,3c,6b and Supplementary Fig. 19).

In *in vitro* studies, we have performed flow cytometry analysis to quantify the cell internalization and the imaging performance of Pep-Acy/Glu@AuNRs in living cells. The corresponding results and detailed experiment information have been provided in the revised

version (Please see revised manuscript, page 9, line 2-6 of paragraph 2; Fig. 3d; page 10, line 13-15 of paragraph 1, Supplementary Fig. 17,18; page 16, paragraph 3). Besides, we have further quantified the cell internalization of Pep-Acy/Glu@AuNRs by measuring the intracellular Au content with ICP-MS. The results show that the intracellular uptake of Pep-Acy/Glu@AuNRs in SCC-7 cells was ca. 2.5-fold increases compared with that in glucosamine pre-treated SCC-7 cells or Pep-Acy@AuNRs in SCC-7 cells, indicating the cell internalization of the nanoprobe was obviously enhanced by glycosyl (Please see revised manuscript, page 9, line 6-8 of paragraph 2; Supplementary Fig. 16, page 17, paragraph 2).

In addition, we have performed quantitative *in vivo* fluorescence imaging analysis by counting the number of photo counts per second in the tumor (treated with MMPs inhibitor or NaHCO₃ and without treatment) and normal tissues as a function of time. The corresponding results have been added in the revised version (Please see the revised manuscript, page 10, paragraph 2 - page 11, paragraph 1; Supplementary Fig. 19). Furthermore, the biodistribution of Pep-Acy/Glu@AuNRs determined by ICP-MS has been supplemented as well (Please see the revised manuscript, page 11, Paragraph 2; page 18, first paragraph; Supplementary Fig. 20).

Comment 4:

Authors need to provide the MMP-13 specificity of this probe.

Reply and Corresponding Changes:

Good suggestion! In fact, the as-prepared Pep-Acy/Glu@AuNRs has specificity for multiple types of MMPs (e.g. MMP-2, MMP-3, MMP-7, MMP-9 and MMP-13), overexpressed in various malignant tumors. Herein, we chose MMP-13 as a typical MMP for *in vitro* studies just because the nanoprobe is much more sensitive to MMP-13. The corresponding experimental data (The responses of the fluorescence spectra of Pep-Acy/Glu@AuNRs to various types of MMPs) and explanations have been supplemented in the revised version (Please see the revised manuscript, page 8, first paragraph; Supplementary Fig. 12). In addition, the MMPs specificity of this nanoprobe has been fully verified via *in vitro* enzyme activation, cells imaging and *in vivo* fluorescence imaging (Please see revised manuscript, page 8, first paragraph - page 11, paragraph 2; Fig. 3a,3d-3f, Fig. 4; Supplementary Fig. 12-14,17-20 and the corresponding statements).

Comment 5:

MMPs are known as secreted enzymes. Should be correlated with MMP protein expressions by western blotting in both cell lines (SCC7 and 293T).

Reply and Corresponding Changes:

Very valuable suggestion! We have evaluated the expression levels of MMPs (including MMP-2, MMP-9 and MMP-13, high specificity for Pep-Acy/Glu@AuNRs) in both SCC-7 and 293T cells by Western blotting and ELISA. The results demonstrate that the expression levels of MMPs in SCC-7 cells are much higher than that of in 293T cells (Please see revised manuscript, page 9, line 7-9; Supplementary Fig. 15).

Comment 6:

In Figure 3, there are no scale bars in cellular images. And activated fluorescence signals need to be correlated with not only fluorescence intensity but also upregulated MMP-13 at protein level, e.g. by western blotting.

Reply and Corresponding Changes:

Thanks a lot for your suggestions. We have added the scale bars in cellular images (Please see revised manuscript, Fig. 3e,3f). In addition, in view of your advice, we have investigated the *in vitro* fluorescence activation of Pep-Acy/Glu@AuNRs towards different concentration of MMP-13. The results show that the fluorescence signal activation of Pep-Acy/Glu@AuNRs is closely correlated with the concentration of MMP-13. The above information has been added now (Please see revised manuscript, page 8, line 9-12 of paragraph 1; Supplementary Fig. 13). Moreover, to verify the fluorescence activation of Pep-Acy/Glu@AuNRs in living cells is also related to the upregulated MMPs level, we have evaluated the expression levels of MMPs including MMP-2, MMP-9 and MMP-13 in 293T and SCC-7 cells treated with or without MMPs inhibitor by Western blotting. The corresponding results and explanations have been supplemented (Please see the revised manuscript, page 9, line 6-9 and page 10, line 4-8; Fig. 3f; Supplementary Fig. 15,17,18).

Response to Reviewer 2

Comment :

Authors in this manuscript describe the synthesis, characterisation and evaluation at *in vitro* and *in vivo* levels of a novel dual-stimuli nanoprobe. The idea for tumor-targeting and fluorescence-guided photothermal therapy is a question that authors addresses using the new nanoprobe. The data provided could be useful for further investigation in the field. There are several points in the text referring on the importance of MMPs and how they could contribute in such proposed targeting. Apart from MMPs there are other critical players in tumour microenvironment to be addressed in such a study (for instance uPA, tPA etc). In any case a link between the matrix metalloproteinases and particularly of the types present in the cell lines used and the tumour microenvironment should be considered. For instance, types of MMPs may be activated and other not. Screening of the MMPs will be also useful.

Reply and Corresponding Changes:

Thanks very much for your kind and constructive comments! Besides MMPs, our strategy for the design of dual-stimuli-responsive and reversibly activatable theranostic nanoprobe is easily to extend to other critical players in tumor microenvironment by simply replacing the linker. For example, the use of another peptide sequence, NH₂-GGRRGGC-SH instead of H₂N-GPLGVRGC-SH, as the linker between Acy and AuNRs enables the fabrication of a nanoprobe (Pep₂-Acy@AuNRs) for specific response to cathepsin B (CtsB), a tumor-specific enzyme overexpressed in various malignant tumors, and the specific fluorescence activation of Pep₂-Acy@AuNRs synergistically governed by CtsB and pH. The above information has been added in the revised version (Please see revised manuscript, page 14, last paragraph - page 15, first paragraph) and the characterization, pH/enzyme-triggered fluorescence activation and the cells imaging of Pep₂-Acy@AuNRs have also been supplemented in Supplementary Information (Please see revised Supplementary Information, Supplementary Fig. 23-28).

The responses of the fluorescence spectra of Pep-Acy/Glu@AuNRs to various types of MMPs including MMP-2, MMP-3, MMP-7, MMP-9 and MMP-13, overexpressed in various malignant tumors have been studied (Please see the revised manuscript, page 8, line 1-10; revised Supplementary Information, Supplementary Fig. 12). The results show that the as-prepared Pep-Acy/Glu@AuNRs has specificity to multiple types of MMPs, in particular, MMP-13. We chose MMP-13 as a typical MMP for *in vitro* studies due to its higher sensitivity than other MMPs. In the revised manuscript, the *in vitro* fluorescence activation of Pep-Acy/Glu@AuNRs towards

different concentration of MMP-13 has been studied as well. The above experimental data and corresponding statements have been added now (Please see the revised manuscript, page 8, line 10-12; Supplementary Fig. 13).

Response to Reviewer 3

The authors reported synthesis and surface modification of a dual stimuli-responsive nanoprobe based on asymmetric cyanine and glycosyl functionalized gold nanorods (AuNRs) for both photo-thermal therapy and tumor targeting. The conjugated glucosamine was aimed to enhance both biocompatibility and tumor targeting capability of the nanoprobe while asymmetric cyanine was utilized as the tumor specific imaging probe and photo-thermal agent. The efficacy of the nanoprobe was examined using in vitro cellular experiments and in vivo animal models. I would publish this interesting paper after major revision noted below:

Comment 1:

The advantages and disadvantages of the previously developed nanoprobe in the field should be mentioned in the introduction part. In the current form, the introduction is too general and does not contain key references and the last developments in the field.

Reply and Corresponding Changes:

Thanks very much for your kind and valuable suggestion! In view of your suggestion, the introduction has been revised. The advantages and disadvantages of the previously developed activatable multifunctional nanoprobe along with several key references has been added now. The major modification and supplement are listed as follows: "To this goal, great efforts have been devoted to conjugate various imaging techniques with therapy agents to realize imaging-guided precision therapy. In particular, activatable imaging modality has received increasing attention in the fabrication of theranostic agents owing to its high specificity and sensitivity. Recently, several multifunctional nanocomposites have been developed to make activatable imaging-guided therapy possible with amplified imaging signals, demonstrating the unique superiority and the great potential of activatable imaging strategy for precision imaging-guided therapy. However, to our knowledge, the signal activation in previous theranostic studies is exclusively irreversible, leading to poor signal-to-noise ratio or even 'false positive'

results as the activated signals are ‘always on.’” (Please see the revised manuscript, page 3, line 12-21; ref. 14-25).

Comment 2:

Many of the unnecessary information (e.g., Figures 2b-d, 3b, 5, and 6b) can be transferred to the SI.

Reply and Corresponding Changes:

Thanks for your good suggestion. In light of your suggestion, Figures 2b-d, 3b, and 6b in original manuscript have been transferred to revised Supplementary Information as Supplementary Fig. 7, Supplementary Fig. 14 and Supplementary Fig. 22 (Please see revised manuscript, Fig. 2, 3 and 6; revised Supplementary Information).

Comment 3:

Figure 1 is not easy to understand and needs to be restructured.

Reply and Corresponding Changes:

Thank you very much for your comment. Figure 1 has been restructured and additional explanation has been provided for clarity (Please see revised manuscript, Page 14, line 2-17; Fig. 1).

Comment 4:

It remains unknown in the text why the authors chose the peptide sequence as the linker.

Reply and Corresponding Changes:

Good question! Matrix metalloproteinases (MMPs), overexpressed in cancer area, not only have distinct roles in tumor invasiveness, metastasis and angiogenesis, but also affect multiple signaling pathways in tumor microenvironment. Hence, in our study, MMPs was chosen as a target, a peptide sequence, H₂N-GPLGVRGC-SH (PLGVR is the cleavable site), specifically hydrolyzable by multiple types of MMPs, was employed as the linker to realize tumor microenvironment specific activatable Förster resonance energy transfer (FRET) from the asymmetric cyanine to the AuNRs (Please see revise manuscript, page 4, line 8-10; page 5, line 13-16).

Comment 5:

The authors claimed that the conjugation of strong hydrophilic moieties increased the hydrodynamic size of NRs from 58 nm to 156 nm. However, the substantial reduction of the zeta potential from +66.4 into -1.1 (i.e. close to isoelectric point) suggests the formation of NRs' aggregates is responsible for the substantial increase in the size of NPs. The broad size distribution for Prep-Acy/Glu@AuNRs, in Fig. 2b, could also confirm the formation of the aggregates. This statement should be revised. Other approaches (e.g., differential centrifugal sedimentation) should be used to define the main reason of the size variations.

Reply and Corresponding Changes:

Thanks very much for your criticisms. In fact, the extinction coefficient and the maximum absorption wavelength of the VU-vis-NIR absorption spectra of the plasmonic nanomaterials are known as the most efficient indicators of their stable dispersion in solution (e.g. *Annu. Rev. Phys. Chem.* **58**, 267-297 (2007); *Chem. Rev.* **107**, 4797-4862 (2007); *Anal. Chem.* **85**, 6580-6586 (2013) and *Annu. Rev. Phys. Chem.* **60**, 167-192 (2009)). Besides, some pervious reports have reported that the aggregation of AuNRs would lead to the change of absorption intensity and the longitudinal absorption position of AuNRs (e.g. *Nano Lett.* **9**, 1651-1658 (2009); *ACS Appl. Mater. Interfaces* **5**, 4076-4085 (2013) and *ACS Appl. Mater. Interfaces* **6**, 5657-5668 (2014)). While in our study, no remarkable changes were observed in both absorption intensity and the longitudinal absorption peak position of the VU-vis-NIR spectra of Pep-Acy/Glu@AuNRs compared with those of AuNRs. Besides, the as-prepared Pep-Acy/Glu@AuNRs presents well-dispersed homogenous core-shell structure in the TEM images. Thus, we conclude that no detectable aggregation happened after conjugation of pep-Acy and Glu. The corresponding statements have been added (Please see revised manuscript, Page 6, line 10; Page 7, line 1-4). According to your suggestion, we have re-determined the average hydrodynamic diameter of Pep-Acy/Glu@AuNRs after differential centrifugal sedimentation, and the improved results have been provided in the revised manuscript (Please see the revised Supplementary Information, Page S7, Supplementary Fig. 7a).

Moreover, we have also given the following explanation to address the question. This significant increase of hydrodynamic size is due to the large hydrodynamic volume of glycosyl (formed thick solvent layer). Since DLS measurements use Brownian movement of the solvated

particle, and assumes that the particle is spherical in shape and covered with a solvent shell. These phenomena were previously reported (e.g. *Drug Dev. Res.* **64**, 114-128 (2005); *Angew. Chem. Int. Ed.* **50**, 9348-9351 (2011); *Nat. Mater.* **13**, 418-426 (2014); *ACS Nano* **5**, 854-862 (2011) and *Small* **11**, 2696-2704 (2015)). Accordingly, the statement of 'Conjugation of strong hydrophilic Pep-Acy and Glu-SH remarkably increased the hydrodynamic size of the AuNRs from 58 nm to 156 nm' in original manuscript has been revised to 'Conjugation of Pep-Acy and Glu-SH remarkably increased the hydrodynamic size of the AuNRs from 58 nm to 149 nm due to the large hydrodynamic volume of glycosyl⁴⁴' (Please see the revised manuscript, Page 6, line 13-15).

Comment 6:

The absorption spectrum of CTAB@AuNRs should be inserted in Fig.2e for a better comparison.

Reply and Corresponding Changes:

Good suggestion! The absorption spectrum of CTAB@AuNRs has been inserted in Fig. 2b in the revised version (i.e. Fig. 2e in original version) (Please see the revised manuscript, Fig 2b).

Comment 7:

With regards to the cytotoxicity experiments: the formation of aggregates can significantly influence the mechanism of cell internalization and the number of NRs inside the cells. This can also affect their biological identity and change the biological efficacy of these NPs. The uptake mechanism (after considering comment 5), should be defined in the revised version. Besides, I strongly recommend that the authors quantitatively compare the uptake of CTAB@AuNRs and Acy/Glu@AuNRs, to provide more tangible evidence on the substantial enhancement of NRs' biocompatibility.

Reply and Corresponding Changes:

Thanks very much for your kind and valuable advice. Yes! The formation of aggregates can significantly influence the cell internalization and the biological efficacy of NPs. Taken into account the comment 5, we have investigated the stability and the dispersion of Pep-Acy/Glu@AuNRs in a simulated physiological medium, Dulbecco's modified Eagle's medium (DMEM) with 10% fetal bovine serum (FBS). No remarkable variation in the absorption intensity and peak position were observed in the UV-vis-NIR absorption spectra. Meanwhile, neither

morphological change nor aggregation appeared in the TEM images. The results show that the prepared Pep-Acy/Glu@AuNRs are stable and well-dispersed in a complex physiological medium. Thus, no effect on the uptake mechanism and the biological efficacy is expected. The corresponding statements and results have been now added (Please see revised manuscript, page 7, line 9-14; Supplementary Fig. 9a,b).

In addition, in view of your valuable advice, we have performed the quantitative analysis of the intracellular uptake of Pep-Acy/Glu@AuNRs and Pep-Acy@AuNRs by ICP-MS and flow cytometry analysis. The mean fluorescence intensity (measured by flow cytometry analysis) of SCC-7 cells treated with Pep-Acy/Glu@AuNRs at pH 6.0 is ca. 2.7-fold increase compared with that of SCC-7 cells incubated with Pep-Acy@AuNRs although the same concentration of nanoprobe (as Au) was applied. Besides, the intracellular Au content of the SCC-7 cells treated with Pep-Acy/Glu@AuNRs also shows ca. 2.5-fold increase compared with that of Pep-Acy@AuNRs. The above results indicate that Pep-Acy/Glu@AuNRs with good biocompatibility was successfully internalized into the SCC-7 cells and the cell internalization was obviously enhanced by glycosyl. The above results have been added in the revised version (Please see revised manuscript, Page 9, last paragraph; Fig. 3d; Supplementary Fig. 16). However, quantitative analysis of intracellular uptake of CTAB@AuNRs is impossible due to its remarkable cytotoxicity. Obvious apoptosis occurred when cells treated with CTAB@AuNRs. It is therefore difficult to quantitatively evaluate the intracellular uptake of CTAB@AuNRs.

Reviewers' Comments:

Reviewer #1 (Remarks to the Author)

I'm generally satisfied with the reply of the authors.

Reviewer #2 (Remarks to the Author)

Authors addressed the comments of this reviewer and improved the quality of the manuscript.

Reviewer #3 (Remarks to the Author)

All of my comments have been carefully considered by the authors in their revised version. I would publish the paper as is.

POINT-BY-POINT RESPONSE TO REVIEWERS

The positive comments made by the reviewers give us a great encouragement. We highly appreciate the reviewers for such kind comments. The referees' comments and our replies are made below.

Reviewer #1 (Remarks to the Author):

I'm generally satisfied with the reply of the authors.

Response: Thanks for your kind comment!

Reviewer #2 (Remarks to the Author):

Authors addressed the comments of this reviewer and improved the quality of the manuscript.

Response: Thanks!

Reviewer #3 (Remarks to the Author):

All of my comments have been carefully considered by the authors in their revised version. I would publish the paper as is.

Response: Thanks very much for your positive comments!